# Extension of the crRNA enhances Cpf1 gene editing in vitro and in vivo

Hyo Min Park[1], Hui Liu[1], Joann Wu[1], Anthony Chong[1], Vanessa Mackley[1], Christof Fellmann [2], Anirudh Rao[2], Fuguo Jiang[2], Hunghao Chu[1], Niren Murthy[3] & Kunwoo Lee [1]

Engineering of the Cpf1 crRNA has the potential to enhance its gene editing efficiency and non-viral delivery to cells. Here, we demonstrate that extending the length of its crRNA at the 5′ end can enhance the gene editing efficiency of Cpf1 both in cells and in vivo. Extending the 5′ end of the crRNA enhances the gene editing efficiency of the Cpf1 RNP to induce non-homologous end-joining and homology-directed repair using electroporation in cells. Additionally, chemical modifications on the extended 5′ end of the crRNA result in enhanced serum stability. Also, extending the 5′ end of the crRNA by 59 nucleotides increases the delivery efficiency of Cpf1 RNP in cells and in vivo cationic delivery vehicles including polymer nanoparticle. Thus, 5′ extension and chemical modification of the Cpf1 crRNA is an effective method for enhancing the gene editing efficiency of Cpf1 and its delivery in vivo.

[1] GenEdit Inc., Berkeley, CA 94720, USA. [2] Department of Molecular and Cell Biology, University of California, Berkeley, Berkeley, CA 94720, USA. [3] Department of Bioengineering, University of California, Berkeley, Berkeley, CA 94720, USA. Correspondence and requests for materials should be addressed to N.M. (email: nmurthy@berkeley.edu) or to K.L. (email: lee@genedit.com)

Class 2 CRISPR (clustered regularly interspaced short palindromic repeats)-encoded Cas effector proteins are RNA-guided endonucleases that can be programmed to cleave DNA targets[1–5]. They have been broadly utilized to edit the genomes of various organisms for both biotechnology and medical purposes[6–9]. Among class 2 proteins, *Streptococcus pyogenes* Cas9 (SpCas9) has been the most actively investigated. SpCas9 has been extensively engineered, optimized, and delivered both in vitro and in vivo using a variety of different modalities and methods[10–16]. By contrast, fewer optimizations have been accomplished for the more recently discovered CRISPR-Cpf1.

Cpf1 proteins from the *Acidaminococcus* sp. BV3L6 (As), *Lachnospiraceae bacterium* (Lb), and *Francisella novicida U112* (Fn) organisms have several innate features that make them attractive alternatives to SpCas9[17–20]. First, Cpf1 has a unique TTTV protospacer adjacent motif (PAM) recognition sequence that expands genomic targeting beyond the guanosine-rich sequences recognized by SpCas9 (NGG PAM)[17,21–24]. Second, Cpf1 possesses an innate RNase activity that has been demonstrated to facilitate the delivery of multiple CRISPR RNAs (crRNAs) as a single-guide RNA (sgRNA)[25–28]. Third, Cpf1 proteins utilize a single crRNA (about 41 nucleotides)[17], which is much shorter than the 100-nucleotide-long crRNA-tracrRNA chimera (sgRNA) used in SpCas9[29,30]. The smaller size of the Cpf1 crRNA facilitates the chemical synthesis and, thereby, the chemical modification of the guide RNA[31]. Despite these advantages, Cpf1 use in research and therapeutic settings is limited. This may be due to the nuclease activity of Cpf1 or the challenges associated with delivering Cpf1 in vitro and in vivo.

Engineering the crRNA of Cpf1 has great potential to enhance both its gene editing efficiency and non-viral delivery to cells. The sgRNA of SpCas9 has undergone extensive sequence, length, and chemical optimizations to enhance gene editing activity[10,13,16,32–34]. In contrast Cpf1 crRNA engineering remains to be intensively explored, although a few studies have demonstrated that modifications at the 3′ end can improve Cpf1 activity[18,31].

We hypothesized that increasing the length of the crRNA scaffold at the 5′ end can enhance the AsCpf1 RNP gene editing and delivery. We selected the 5′ end for crRNA engineering because various AsCpf1–crRNA complex structures show the 5′ terminal of the crRNA scaffold to be largely exposed and potentially suitable for engineering[24,36]. Also, it is unclear if the biochemically identified minimal crRNA scaffold is the optimal crRNA scaffold for Cpf1-mediated gene editing in eukaryotic systems[17,25,28], and whether extending the 5′ end could enhance editing efficiency. Lengthening the crRNA scaffold at the 5′ end may also enhance the delivery of the AsCpf1 RNP using cationic materials by increasing the complex's overall negative charge density.

Here, we demonstrate that extending the 5′ end of the crRNA increases both the editing efficiency and delivery of AsCpf1 in vitro and in vivo. First, we show that a 2 to 59 nucleotide extension to the 5′ end significantly increases AsCpf1-mediated editing in electroporated cells. This enhancement is robust and occurs in both immortalized and primary cells. Second, we demonstrate that short 5′ extensions increase the tolerance of the crRNA 5′ end to chemical modifications, which results in enhanced serum stability. Finally, we show that AsCpf1 complexed with a crRNA with a 59 nucleotide extension to the 5′ end has dramatically increased gene editing efficiency both in vitro and in vivo, after delivery with cationic delivery vehicles (Fig. 1).

## Results and discussion
**Extending the crRNA 5′ end increases Cpf1 RNP gene editing**. SpCas9 RNP exhibits robust gene editing in cells[37–41]. To assess whether AsCpf1 RNP is also capable of achieving high gene editing levels, we compared the two RNPs using a green fluorescent protein (GFP) reporter system. We selected a matched protospacer sequence in GFP that could be recognized by both AsCpf1 and SpCas9 in order to directly compare the two nucleases (Fig. 2a). The RNP complexes were introduced into HEK293T cells expressing the GFP gene under the control of a doxycycline-inducible promoter (GFP-HEK) using electroporation. Editing activity was determined by measuring the population of GFP-negative cells, with GFP expression disrupted through non-homologous end-joining (NHEJ)-mediated indel mutations. AsCpf1 RNP exhibited lower gene editing than SpCas9 in the electroporated cells (Fig. 2b).

We performed experiments to determine if having additional nucleotides on the 5′ end of the Cpf1 crRNA could enhance its gene editing efficiency. Engineering of the sgRNA scaffold has been shown to enhance SpCas9 gene editing efficiencies[13]. However, analogous investigations have not been conducted for Cpf1. To determine if crRNA 5′ extensions affect Cpf1 gene editing, we compared the activities of crRNAs with 5′ extensions of various lengths using our GFP-HEK reporter system. GFP-targeting crRNAs with 5′-end extensions of 4, 9, 15, 25, and 59 nucleotides were introduced into GFP-HEK cells by electroporation as an RNP complex with AsCpf1. The sequences for the 4 to 25 nucleotide extensions were scrambled, and the 59 nucleotide extension consisted of the AsCpf1 pre-crRNA[17,25] preceded by a scrambled RNA sequence with no homology to human genomic sequences. The crRNAs with the 4 to 25 nucleotide 5′ extensions all exhibited dramatically increased gene editing over the crRNA with no extension. Cells electroporated with the unextended crRNA were 30% GFP negative (crRNA), 4 to 25 nucleotide extended crRNA were 55 to 60% GFP negative and 59 nucleotide extended crRNA were 37% GFP negative (crRNA$^{+59}$) (Fig. 2c). The gene editing levels for the 4 and 25 nucleotide 5′ extended crRNAs are comparable to that of the SpCas9 RNP-electroporated cells (Fig. 2b).

**5′ Extended crRNAs increase the HDR levels of AsCpf1**. We also examined whether the 5′ extension could increase homology-directed recombination (HDR) rates in addition to NHEJ levels. The AsCpf1 RNPs with crRNA containing various extensions were introduced into GFP-HEK cells together with a single-stranded oligonucleotide donor (ssODN) (Fig. 3a). HDR rates were quantified using a restriction enzyme digestion assay[40,42–44]. A twofold improvement in HDR was observed for both the 4 and 9 nucleotide extended crRNAs (17% HDR frequency for crRNA$^{+4}$ and 18% HDR frequency for crRNA$^{+9}$ vs. 9% for control crRNA in Fig. 3b).

Interestingly, cells treated with ssODN + Cpf1 RNP also had a dramatic increase in the number of GFP-negative cells. The GFP-negative cells generated from ssODN + Cpf1 RNP were caused by frameshift mutations due to HDR and indel mutations caused by NHEJ. The ssODN increased the percentage of GFP-negative cells, which includes both NHEJ and HDR populations, from 30 to 46% for the unextended crRNA (crRNA), 55 to 92% for crRNA$^{+4}$, 58 to 90% for crRNA$^{+9}$, and 37 to 58% for crRNA$^{+59}$ (Fig. 3c). We performed additional experiments to determine if the exogenously added DNA had to have homology to the Cpf1 RNP target site in order to enhance gene editing. AsCpf1 RNP was electroporated into cells along with single-stranded DNA (ssDNA) without any homology to the target sequence, and the gene editing efficiency was measured. Similarly, the addition of ssDNA without homology also increased the AsCpf1 editing activity to approximately 90% for both extended crRNAs (Fig. 3d). ssDNA can augment the editing efficiency of SpCas9[45], delivered

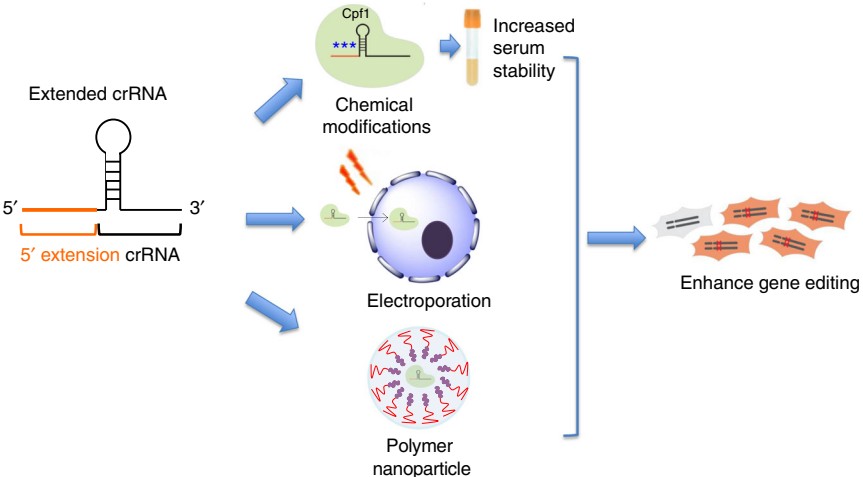

**Fig. 1** crRNA with a 5′ extension enhances the gene editing efficiency and delivery of Cpf1. Extending the length of the Cpf1 crRNA at its 5′ end improves the biological performance of the Cpf1 RNP. crRNA with a 5′ extension tolerates chemical modifications, and this results in greater serum stability. Cpf1 RNP with a 5′ extension has increased gene editing (both NHEJ and HDR) in cells after delivery via electroporation. In addition, Cpf1 RNP with a 5′ extension is more efficiently delivered into cells and in vivo with cationic lipids and polymer

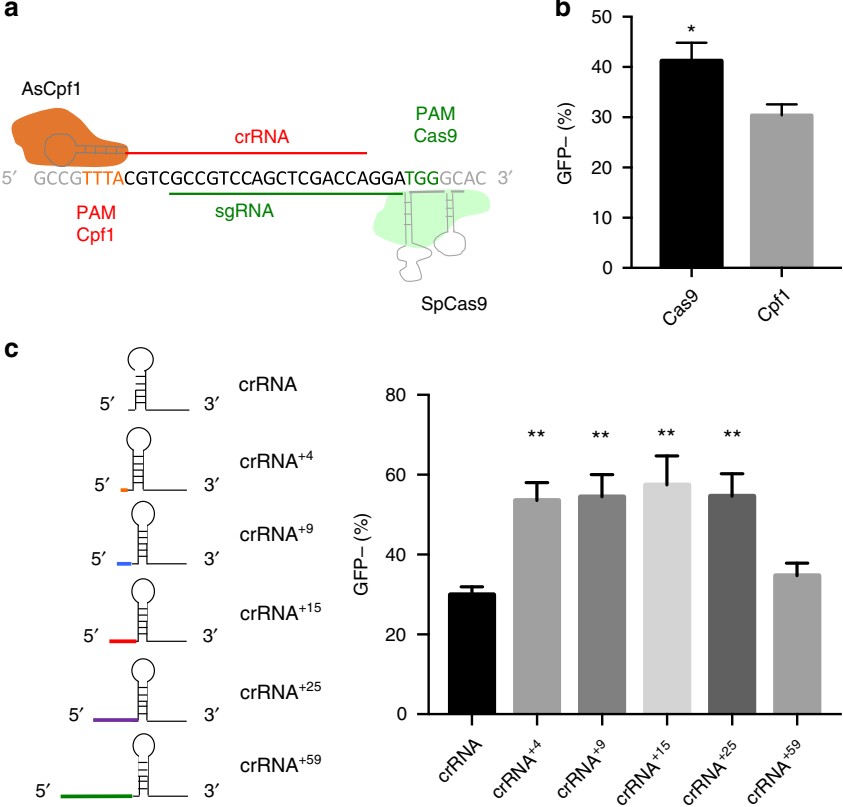

**Fig. 2** 5′-End extensions of the crRNA enhance the gene editing of Cpf1 in HEK cells. **a** Sequence of the GFP protospacer that is targeted by both AsCpf1 and SpCas9. **b** Electroporation of SpCas9 and AsCfp1 RNP targeting the GFP-matched site demonstrate that SpCas9 can knock out genes more efficiently than Cpf1. Mean ± SE, $n = 4$. *$p < 0.05$ by Student's $t$ test. **c** 5′ Sequence extension of crRNA increases Cpf1 gene editing in GFP-HEK cells. Cpf1 RNP+ with various 5′ extended crRNAs were delivered to GFP-HEK cells using electroporation. Four nucleotide to twenty-five nucleotide extension significantly increased editing efficiency. Mean ± SE, $n = 3$. **$p < 0.01$ by Student's $t$ test compared to crRNA control

via electroporation, and our results demonstrate that ssDNA can augment editing with AsCpf1 as well. Additionally, the activity enhancement of the 5′-end extension was synergistic with exogenously added ssDNA and collectively the gene editing they induced was close to a 100%, which is a level that had not been reported previously. We also tested whether ssDNA could increase the gene editing efficiency of Cpf1 after delivery into cells via Lipofectamine. Supplementary Figure 1 shows that the addition of ssDNA does not enhance the Cpf1 gene editing efficiency, if Lipofectamine is used as the delivery method. This

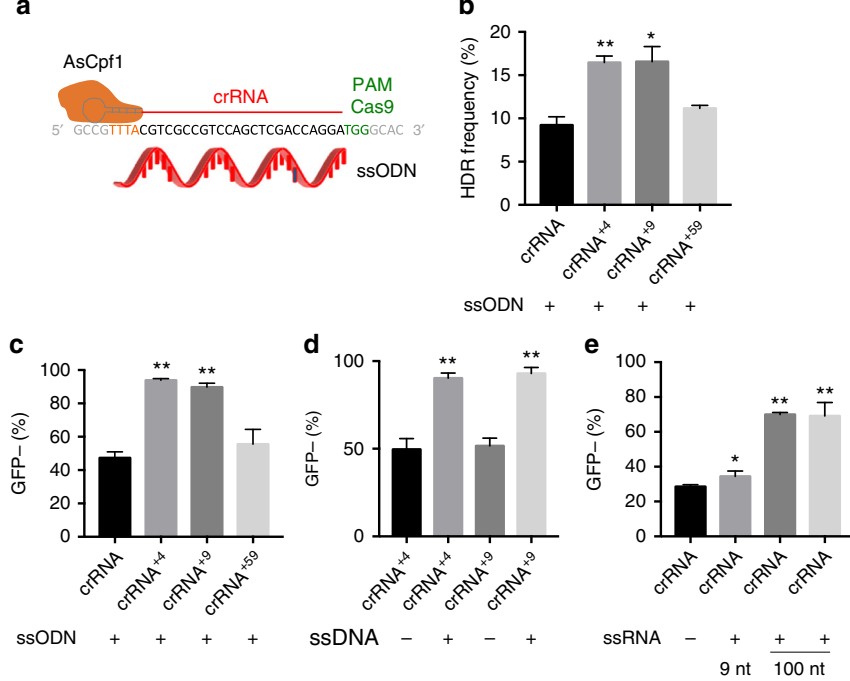

**Fig. 3** Single-stranded DNA has synergistic effects with 5′ extended crRNAs and enhances both NHEJ and HDR in HEK cells. **a** Cpf1 RNP and ssODN were electroporated into cells and the HDR and NHEJ rates were determined. **b** The HDR frequency in GFP-HEK cells is enhanced with Cpf1 RNP+. The donor ssODN contained a restriction enzyme site, which was used for a restriction enzyme digestion assay to determine the HDR frequency. Mean ± SE, $n = 3$. *$p$ < 0.05, **$p$ < 0.01 by Student's $t$ test compared to crRNA control. **c** GFP knockout efficiency of Cpf1 RNP+ was further improved in GFP-HEK cells when ssODN was delivered together with electroporation. Cpf1 RNP+ with various 5′ extended crRNAs and donor ssODN were delivered to GFP-HEK cells using electroporation. Mean ± SE, $n = 3$. **$p$ < 0.01 by Student's $t$ test compared to crRNA control. **d** The GFP knockout efficiency of Cpf1 RNP+ was improved in GFP-HEK cells when ssDNA was exogenously added. Cpf1 RNP+ with various 5′ extended crRNAs and random 100 nt ssDNA without homology to the target sequence were delivered to GFP-HEK cells using electroporation. ssDNA enhanced gene editing efficiency of Cpf1 RNP close to 90%. Mean ± SE, $n = 3$. **$p$ < 0.01 by Student's $t$ test compared to each without ssDNA control. **e** ssRNA enhances the knockout efficiency of Cpf1 RNP in GFP-HEK cells. Cpf1 RNP and random ssRNAs (without sequence similarity) were delivered to GFP-HEK cells using electroporation. Mean ± SE, $n = 3$. *$p$ < 0.05, **$p$ < 0.01 by Student's $t$ test compared to crRNA control

result limits the usage of ssDNA as an enhancer of gene editing to the electroporation method.

We further investigated whether single-stranded RNA (ssRNA) can enhance the gene editing efficiency of AsCpf1. ssDNA is potentially problematic to use for enhancing AsCpf1 gene editing activity because ssDNA can potentially integrate into the genome and cause genomic damage; in contrast, ssRNA cannot integrate into the genome, and would be much safer to use. GFP-HEK cells were electroporated with Cpf1 RNP and two different ssRNAs (9 and 100 nucleotides) and the resulting levels of gene editing were determined. Two 100 nucleotide ssRNAs with slight sequence variation both dramatically increased the gene editing efficiency of Cpf1, resulting in a twofold improvement, whereas the 9 nucleotide ssRNA induced only a 10% enhancement in gene editing efficiency (Fig. 3e). These results suggest that 100 nucleotide ssRNA can be potentially used as a gene editing enhancer for Cpf1 RNP, and provides a safe alternative to ssDNA.

**Extended crRNAs enhance editing efficiency in primary cells.**
We performed gene editing experiments with Cpf1 RNP complexed to 5′ extended crRNAs, in primary mouse myoblasts, to determine if the gene editing enhancements seen in GFP-HEK cells could be applied to other cell types. We choose primary mouse myoblasts as a second test bed because of their importance in treating genetic muscular dystrophies, including Duchenne muscular dystrophy. Primary myoblasts isolated from the Ai9 mouse (ai9 myoblasts)[46,47] were electroporated with AsCpf1 RNP

complexed with crRNAs with and without 5′ extensions, and resulting levels of gene editing were determined. The Ai9 mouse is a transgenic mouse strain, which contains a tdTomato reporter gene that has stop codons in all three reading frames coupled to a triple poly(A) signal (Fig. 4a). The AsCpf1 spacers were designed to introduce multiple breaks into the reporter gene, which would remove the stop sequences by deletion, allowing for genetic editing to be monitored via the expression of tdTomato (a red fluorescent protein, RFP), either through fluorescence microscopy (Fig. 4a) or flow cytometry (Fig. 4b).

The extended crRNAs showed a 25–60% increased gene editing over the unextended crRNA in primary myoblasts. Myoblasts treated with unextended crRNA were 12% RFP positive, 2 nucleotide extended crRNA were 15% RFP positive, 9 nucleotide extended crRNA were 20% RFP positive, and 59 nucleotide extended crRNA were 18% RFP positive (Fig. 4b). Additionally, the effects of ssDNA or ssRNA were also tested in primary myoblasts, and both ssDNA and ssRNA (100 nucleotide length) enhanced the gene editing efficiency of Cpf1 RNP in primary myoblasts (Fig. 4c).

Finally, we investigated if 5′ extended crRNAs could enhance the ability of the Cpf1 RNP to edit an endogenous gene, *SERPINA1*, as a testbed. *SERPINA1* was selected for further investigation because mutations in the *SERPINA1* gene cause α1-anti-trypsin deficiency[48,49], which makes it a target for therapeutic gene editing. Cpf1 with either crRNA or crRNA[+9], targeting the *SERPINA1* gene, were transfected into HepG2 cells via electroporation. Figure 4d shows that Cpf1 RNP with crRNA

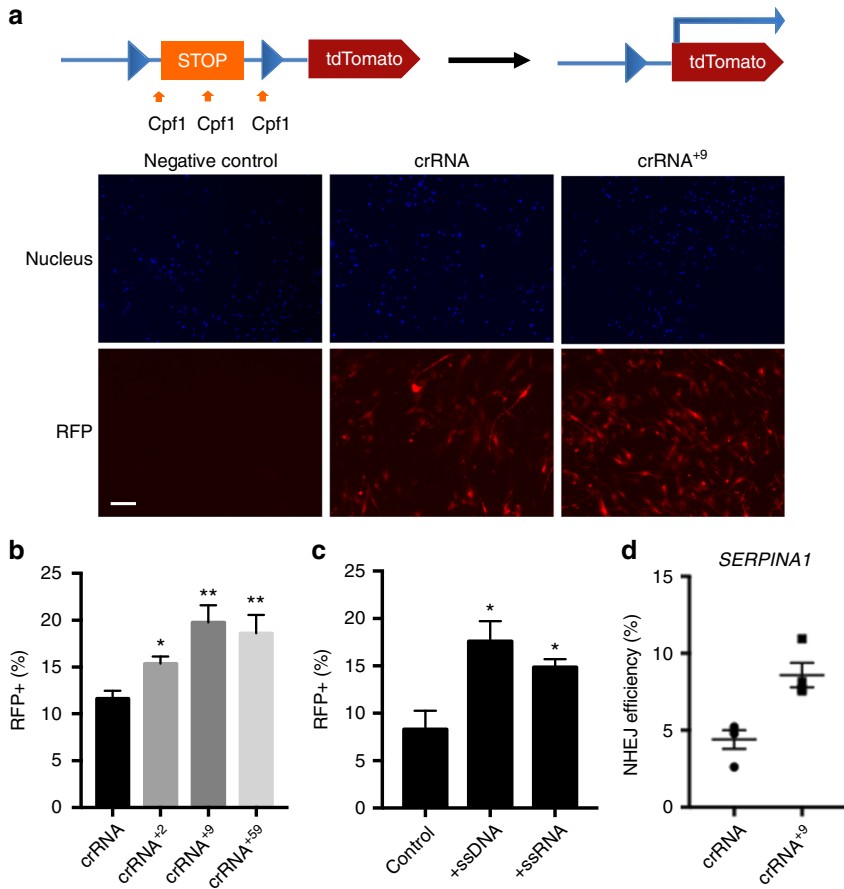

**Fig. 4** The nuclease activity of Cpf1 is enhanced with 5′-terminal extensions in primary myoblasts and HepG2 cells. **a** Fluorescence images of primary ai9 myoblasts after Cpf1 RNP delivery. Negative control had no treatment. Cpf1 RNP and Cpf1 RNP$^{+9}$ were delivered into cells using electroporation. Hoechst staining visualized the nucleus and RFP expression is visualized in red. Scale bar is 100 μm. **b** 5′ Extensions of crRNA enhance Cpf1 gene editing in primary ai9 myoblasts. Quantification of RFP+ resulting from target sequence deletion after electroporation of Cpf1 RNPs with various crRNA 5′ extensions. Mean ± SE, $n = 3$. *$p < 0.05$, **$p < 0.01$ by Student's $t$ test compared to crRNA control. **c** Addition of ssDNA or ssRNA enhances the gene editing efficiency of Cpf1 RNP. Cpf1 RNP was transfected to ai9 myoblasts with either 100 nt ssDNA or 100 nt ssRNA (no sequence homology to target DNA). Quantification of RFP + cells indicates that single-stranded nucleic acids enhance the gene editing efficiency of the Cpf1 RNP. Mean ± SE, $n = 3$. *$p < 0.05$ by Student's $t$ test compared to control. **d** Extended crRNA increases the gene editing efficiency of Cpf1 RNP in HepG2 cells. Cpf1 RNP designed to target the *SERPINA1* gene was delivered into HepG2 cells using electroporation. Droplet digital PCR (ddPCR) was conducted on genomic DNA from the HepG2 cells to quantify NHEJ efficiency. Mean ± SE, $n = 4$, $p = 0.0057$ by Student's $t$ test compared to crRNA control

$^{+9}$ had an enhanced NHEJ efficiency in comparison to wild-type crRNA. Collectively, these results suggest that the enhancing gene editing effects of the 5′ crRNA extensions are broadly applicable across genetic targets and cell types.

**The 5′ crRNA extension tolerates chemical modifications.** Chemical modification of crRNAs and sgRNAs have great potential for improving the gene editing efficiency of CRISPR/Cas proteins. For example, chemical modification of sgRNAs at the 5′ and 3′ end improves the SpCas9 activity in both ex vivo and in vivo settings[10,33,34,50], and chemical modification of Cpf1 crRNA at its 3′ end also results in a modest enhancement in gene editing activity[31]. However, the 5′ end of the Cpf1 crRNA has been intractable to chemical modifications, and modification results in a loss of nuclease activity[31,51]. Thus, it has been challenging to improve the activity of the Cpf1 RNP with chemically modified crRNAs due to their low tolerance towards chemical modifications.

We therefore investigated if extending the crRNA at its 5′ end increases its tolerance to chemical modifications. Three different chemical modifications were investigated on crRNAs with

extension. In particular, we introduced 2′ O-methyl modifications, phosphorothioate linkages, and deoxynucleotide ribose groups to the crRNA extensions (Fig. 5a). We chose these modifications because they have enhanced the stability of various RNAs, including small interfering RNAs (siRNAs) and SpCas9 sgRNAs[10,33,50,52–55]. The chemical modification applied to the 5′ extended crRNA are shown in Fig. 5. Three different chemical modification of the 5′ extended crRNAs were investigated. These were: (1) a crRNA with the first 3 of the 4 nucleotides extended with 2′-O-methyl nucleotides and 3′ phosphorothioate linkage (MS), (2) a crRNA with a deoxynucleotide at the 9th position of the 9 nucleotide 5′ extended crRNA (9dU), and (3) a crRNA with a 3′ phosphorothioate linkage at all 9 nucleotides plus a deoxynucleotide at the 9th position of the 9 nucleotide 5′ extended crRNA (9S). Cpf1 RNP with crRNAs that had extensions and chemical modifications were electroporated into GFP-HEK cells and the gene editing activity was determined by flow cytometry. Extended crRNA with chemical modifications had similar activity to unmodified extended crRNA (41–46% GFP-negative cells) (Fig. 5b). Also, these crRNAs were examined using a blue fluorescent protein (BFP) expressing HEK293T cell line (BFP-HEK). Similar to the above studies with the GFP-HEK

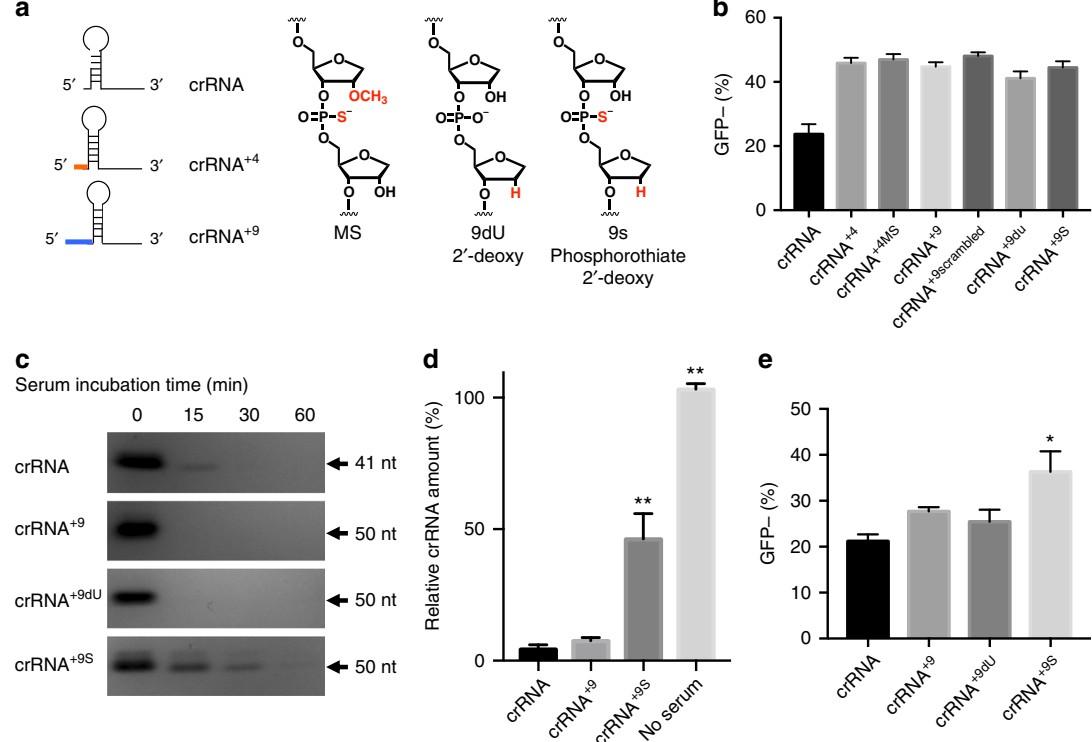

**Fig. 5** Chemical modifications on the 5′ extended crRNA are well tolerated and enhance serum stability. **a** crRNAs with 4 and 9 nt extensions, using a variety of sequences: MS is 2′-OMe 3′-phosphorothioate modifications on the first three nucleotides from the 5′ end, +9dU is 2′-deoxy modification on the 9th nucleotide from the 5′ end, +9S is phosphorothioate modifications on the first 9 nucleotides from the 5′ end on top of 2′-deoxy modification on the 9th nucleotide. **b** Chemical modifications on extended crRNAs are well tolerated and show similar Cpf1 gene editing efficiency to unmodified crRNAs in GFP-HEK cells. Mean ± SE, $n = 3$. All extended crRNAs had a statistically significant difference to unmodified crRNA with a $p$ value less than 0.01 by Student's $t$ test. **c** crRNA with phosphorothioate backbone modification (crRNA$^{+9S}$) has enhanced serum stability. crRNAs with and without chemical modification were incubated with serum and analyzed by gel electrophoresis. **d** Quantification of crRNAs after 15 min incubation in serum. Relative band intensity of crRNA, normalized to crRNA without serum incubation. Mean ± SE, $n = 3$. **$p < 0.01$ by Student's $t$ test compared to crRNA control. **e** crRNA with a 9 nt extension that has been modified with a phosphorothioate backbone (crRNA$^{+9}$) has enhanced Cpf1 gene editing efficiency after delivery with Lipofectamine. Cpf1 RNPs with various crRNAs were delivered with Lipofectamine to GFP-HEK cells and the GFP-negative population was quantified. Mean ± SE, $n = 3$. *$p < 0.05$ by Student's $t$ test compared to crRNA control

cells, the 5′ extensions increased the gene editing efficiency of AsCpf1 and the tolerance of the 5′ end of the crRNA for chemical modifications (Supplementary Fig. 2). These results demonstrate that 5′ chemical modifications of the crRNA are possible without damaging the activity, if the 5′ end of the crRNA is extended.

A key benefit of using chemically modified crRNAs is that they are more stable to hydrolysis by serum nucleases. Therefore, the serum stability of the 5′ chemically modified crRNAs was investigated. 5′ Chemically modified crRNAs were incubated in diluted fetal bovine serum (FBS) and their degradation was analyzed via gel electrophoresis. Figure 5c, d show that unmodified crRNAs rapidly degrade in serum, whereas crRNA$^{+9S}$, which contains a phosphorothioate backbone, is significantly more stable to hydrolysis in serum. In addition, we investigated if 5′ modified crRNAs could enhance the ability of Lipofectamine to transfect Cpf1 RNP, due to its ability to protect the crRNA from nucleases in cells and in serum. Cpf1 with crRNA$^{+9S}$ was more efficient at editing genes in cells than crRNA$^{+9}$ by 40%, and this suggests that 5′ crRNA chemical modifications, enabled by 5′ crRNA extension, will have numerous applications in gene editing (Fig. 5e).

The crystal structure of Cpf1 RNP has recently been solved and demonstrates that the AsCpf1 protein forms numerous interactions with the phosphodiester backbone of the crRNA[24,36]. 5′ Chemical modifications of the unextended crRNA therefore has a high chance of disrupting important interactions between the

crRNA and the Cpf1, resulting in a disruption of AsCpf1 gene editing activity. In contrast, crRNA with 5′ extensions appear to tolerate chemical modifications because the nucleotides interacting with the AsCpf1 protein are not modified. These results provide a methodology for introducing chemical modifications at the 5′ end of the crRNA, which can potentially enhance Cpf1 delivery for ex vivo and in vivo therapeutic applications[10,31,50].

**crRNA extension enhances delivery to cells by cationic lipid.** Unlike SpCas9 that has been delivered with cationic lipids like Lipofectamine[38,56,57], Cpf1 has not been delivered with cationic delivery vehicles. This may be because of the potentially poor interaction between Lipofectamine and the Cpf1 RNP due to the low negative charge of the RNP complex. In contrast to the SpCas9 sgRNA (100 nt), the Cpf1 crRNA is significantly shorter (41 nt). A shorter RNA may reduce the ability of the Cpf1–crRNA complex to interact with cationic lipids. As crRNA 5′ end tolerates extensions of various lengths (Figs. 2, 3, 4), we examined whether 5′-end extensions enhanced AsCpf1 gene editing in cells using cationic lipids. AsCpf1 RNP complexed with crRNAs containing a 0, 9, or 59 nucleotide extensions were introduced into GFP-HEK cells using Lipofectamine 2000. Both the 9 and 59 nucleotide extended crRNAs exhibited increased gene editing over the unextended crRNA: unextended crRNA cells were 8%

GFP negative (crRNA), the 9 nucleotide extended crRNA cells were 18% GFP negative (crRNA$^{+9}$), and the 59 nucleotide extended crRNA cells were 37% GFP negative (crRNA$^{+59}$) (Fig. 6). In addition, crRNA$^{+9}$, crRNA$^{+15}$, and crRNA$^{+25}$ were tested with Lipofectamine and increased the Cpf1 gene editing efficiency in a length-dependent manner (Supplementary Fig. 3).

Next, to ascertain if there is a specific 5′-extension sequence requirement for this enhancement, three different 59 nucleotide 5′ extensions were compared. The first and original 59 nucleotide extended crRNA is described above and contains one AsCpf1 pre-crRNA (crRNA$^{+59}$), the second 59 nucleotide extended crRNA contains four AsCpf1 pre-crRNA sites in tandem (crRNA

$^{+59}$-D2), and the third 59 nucleotide extended crRNA contains the FnCpf1 pre-crRNA[17,28,58] preceded by a scrambled DNA sequence with no homology to any sequence in the human genome (crRNA$^{+59}$-D3) (Fig. 6c). RNP complexes with these crRNAs were delivered using Lipofectamine 2000. All three 5′ extensions showed equivalent editing activity: crRNA$^{+59}$ cells were 32% GFP negative, crRNA$^{+59}$-D2 cells were 30% GFP negative, and crRNA$^{+59}$-D3 cells were 27% GFP negative (Fig. 6c). This suggests that there is not a stringent sequence requirement for the 5′-extension enhancement, similar to previous findings with the 9 nucleotide extended crRNAs with electroporated cells (Fig. 2). Additionally, these results provide

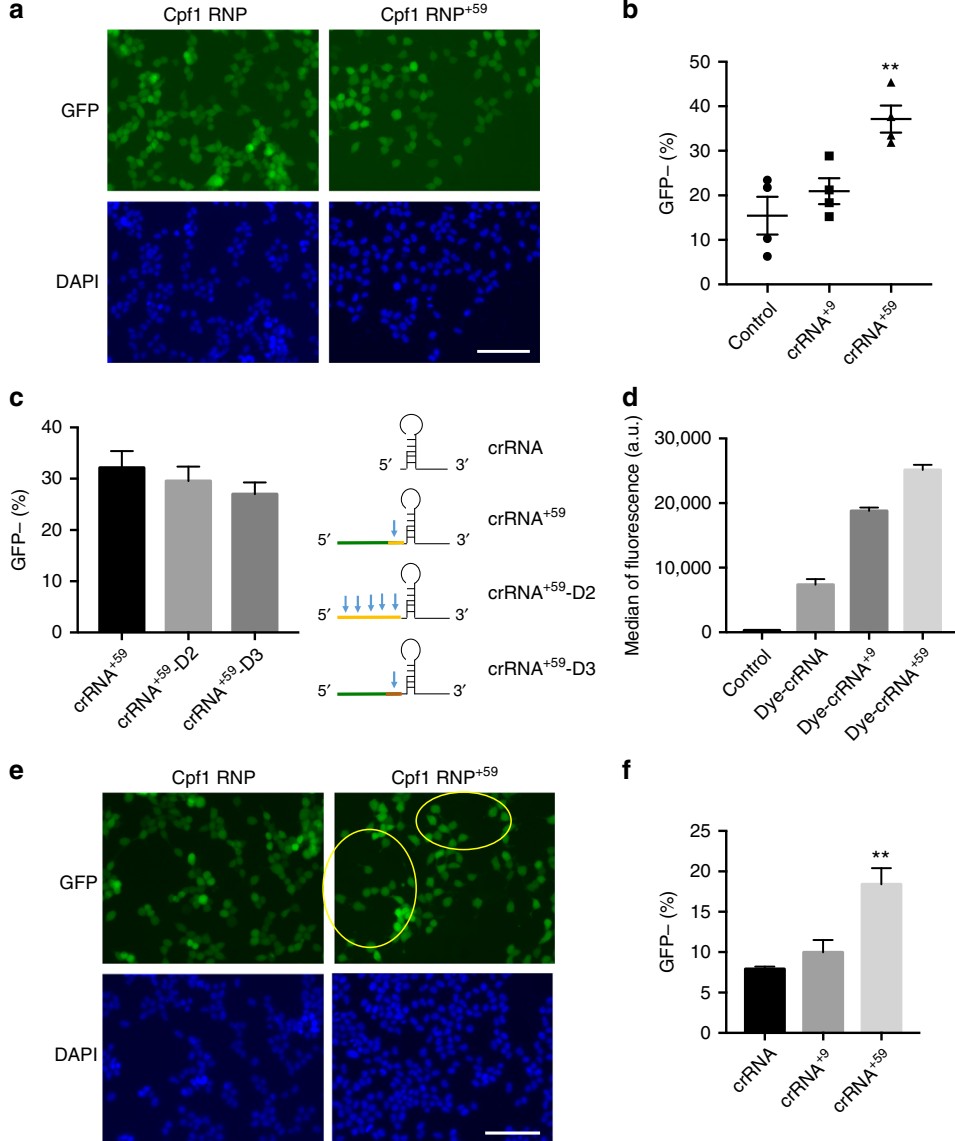

**Fig. 6** crRNA$^{+59}$ enhances the delivery efficiency of Cpf1 RNP with Lipofectamine or cationic polymers in GFP-HEK cells. **a** Fluorescence image of GFP-HEK cells after Cpf1 RNP transfection with Lipofectamine. Hoechst stains nucleus. Scale bar is 100 μm. **b** Quantification of GFP knockout cell population after Cpf1 RNP delivery with Lipofectamine. crRNA$^{+59}$ complexed with Cpf1 (Cpf1 RNP$^{+59}$), 100 nucleotide in length, is more efficient at inducing indel mutations in cells after Lipofectamine transfection than wild-type crRNA complexed with Cpf1 (41 nt in length). Mean ± SE, $n = 3$. **$p < 0.01$ by Student's $t$ test compared to crRNA control. **c** Schematic diagram of crRNA$^{+59}$ sequences with three different sequence extensions. Quantification of GFP knockout cell population after Cpf1 RNP delivery with Lipofectamine. Mean ± SE, $n = 3$. **d** Cellular delivery of Cpf1 RNP using Lipofectamine is dependent on crRNA length. crRNAs with different length were labeled with a fluorescent dye and Cpf1 RNPs were delivered with Lipofectamine in HEK 293T cells. The median fluorescence intensity in each cell was higher with crRNA extension. Mean ± SE, $n = 3$. **e** Fluorescence image of GFP-HEK cells after Cpf1 RNP transfection with PAsp(DET). The areas that cells lost GFP are marked with yellow circles. Hoechst stains nucleus. Scale bar is 100 μm. **f** Quantification of GFP knockout cell population. Cpf1 RNP$^{+59}$, 100 nucleotides in length, is more efficient at inducing NHEJ in cells after PAsp(DET) polymer nanoparticle transfection than wild-type crRNA (41 nt in length). Mean ± SE, $n = 3$. **$p < 0.01$ by Student's $t$ test compared to crRNA control

evidence supporting the above hypothesis that increasing the negative charge density on the crRNA and, thereby, the AsCpf1–RNP complex can enhance the delivery of AsCpf1 to cells by cationic lipids.

We performed experiments to determine the mechanism by which the 5′ extended crRNAs enhanced the gene editing efficiency of the Cpf1 RNP. First, we tested whether the extended crRNAs enhance the inherent nuclease activity of Cpf1, using an in vitro DNA cleavage assay. We observed no activity difference between the three crRNAs tested, wild-type crRNA, crRNA$^{+9}$, and crRNA$^{+59}$ with 15 and 60 min incubation times (Supplementary Figs. 4 and 5). crRNA$^{+59}$ even had slower DNA cleavage than wild-type crRNA when the incubation time was only 5 min. This result suggests that crRNA extension does not enhance the inherent nuclease activity of the Cpf1 RNP. We also investigated if 5′ extended crRNAs enhanced the gene editing activity if the Cpf1 was delivered by plasmid rather than as an RNP. Cpf1 plasmid was transfected 24 h prior to electroporation of the crRNAs and the gene editing activity was determined. Extended crRNAs showed no improvement in gene editing efficiency when the Cpf1 was produced from plasmids (Supplementary Fig. 6).

Finally, we labeled the crRNAs with a fluorescence dye to determine if the extended crRNAs had enhanced uptake in cells after delivery via either electroporation or Lipofectamine. Electroporation of the Cpf1 RNPs resulted in above 90% of the cells being positive for the dye-crRNA and showed highly efficient delivery regardless of the crRNA length. On the other hand, the delivery efficiency of Cpf1 RNP with Lipofectamine was dependent on the length of the crRNA, and extended crRNAs were delivered into HEK 293T cells more efficiently than wild-type crRNA (Fig. 6d). The net charge of a macromolecule is a critical parameter for efficient interaction with Lipofectamine. Extension of the crRNA significantly increases the net negative charge of the Cpf1 RNP, which should result in more efficient interaction with Lipofectamine and efficient delivery into cells.

**crRNA extension enhances Cpf1 delivery by cationic polymers.** Cationic polymers have been investigated for CRISPR delivery because of their ability to efficiently deliver CRISPR components to a variety of cell types and animal models[46,51,59]. Poly(aspartic acid) derivates are cationic polymers with well-established ability to deliver siRNA and the SpCas9 RNP in vivo[46,60,61]. Similarly to cationic lipids, we hypothesized that the 5′ extended crRNA can enhance cationic polymer-based delivery of AsCpf1 RNP to cells. We performed experiments to determine whether the 5′ extended crRNA could also boost the delivery of the AsCpf1 RNP to cells using PAsp(DET). AsCpf1 RNPs with either the 9 or 59 nucleotide 5′ extended crRNA were introduced to GFP-HEK cells using PAsp(DET). Immunofluorescence microscopy and flow cytometry were used to detect the GFP-negative population as surrogate readouts for delivery (Fig. 6e, f, respectively). Similar to the above findings with cationic lipids, the 59 nucleotide 5′ extension also enhanced PAsp(DET)-mediated delivery of AsCpf1 RNP to cells by twofold. The unextended crRNA (crRNA) was 8% GFP negative, the 9 nucleotide extended crRNA (crRNA$^{+9}$) was 10% GFP negative, and the 59 nucleotide extended crRNA (crRNA$^{+59}$) was 18% GFP negative (Fig. 6f). Collectively, the findings from this section, along with previous sections, suggest that the 59 nucleotide 5′ extension of the crRNA is a versatile tool for increasing the delivery of AsCpf1 RNP to cells using cationic materials.

**Efficient in vivo delivery of Cpf1 by polymer nanoparticles.** There is great interest in developing technologies that can safely

and effectively deliver the Cpf1 RNP in vivo. Currently, the only method for delivering Cpf1 in adult mammals is through the use of viruses[25]. However, compared to viral-based methods, direct delivery of the Cpf1 RNP has several advantages because it avoids the immunogenicity problems associated with using viruses[62–66], is straightforward to manufacture[67–70], and potentially generates low levels of off-target DNA damage[35,71]. Non-viral delivery vehicles that can deliver the Cpf1 RNP in vivo therefore have the potential to dramatically accelerate the development of Cpf1 therapeutics.

Since the crRNA$^{+59}$ enhanced delivery and gene editing of AsCpf1 RNP in vitro, we conducted studies to assess whether this extension could also bolster in vivo delivery and editing with either polymer nanoparticles or Lipofectamine. Among a number of poly(aspartic acid) derivative polymers (PAsp analogs) synthesized, one PAsp derivative polymer was identified that could efficiently deliver Cpf1 to myoblasts in vitro and was investigated in vivo. Ai9 mice were given one intramuscular injection of polymer or Lipofectamine combined with either Cpf1 RNP or Cpf1 RNP$^{+59}$. Two weeks after the injection, the expression of tdTomato (red fluorescence) was imaged in 10 μm sections of the gastrocnemius muscle (Fig. 7a). The Cpf1 RNP with a crRNA extension was more efficiently delivered by both Lipofectamine and polymer and efficient gene editing of its target sequence was observed via expression of tdTomato in muscle sections. Additionally, we investigated how broad the region of editing in the muscle was with the Cpf1 RNP$^{+59}$ after delivery with polymer nanoparticles. tdTomato was expressed along the muscle fibers in the gastrocnemius muscle, and the polymer nanoparticle + Cpf1 RNP$^{+59}$ formulation edited an area several millimeters away from the injection site (Fig. 7b).

The extension of the crRNA of Cpf1 allowed efficient nanoparticle formulation with cationic polymers and effective delivery of Cpf1 into the muscle tissue. As a non-viral delivery method, polymer nanoparticles have great clinical potential, because of their low toxicity and low-cost manufacturing process. The enhanced delivery of Cpf1 RNP with polymer nanoparticles in vivo, enabled by crRNA extension, bolsters the value of Cpf1 as a potential therapeutic for treating human diseases.

The crRNA 5′ extensions presented here are straightforward to apply and broadly applicable across cell types. As such, they are powerful tools for developing new gene editing reagents and therapeutics. We demonstrate that extension of the 5′ end of the Cpf1 crRNA increases the gene editing efficiency of Cpf1 RNP in cells. In addition, extending the 5′ end of the Cpf1 crRNA also enables chemically modified bases to be incorporated into the crRNA, which is impossible to do at the 5′ end without a crRNA extension. Finally, extension of the Cpf1 crRNA significantly increases the negative charge density of the Cpf1 RNP and significantly improves the ability of cationic materials to deliver the Cpf1 RNP in vitro and in vivo. We envision that polymer nanoparticle–Cpf1 RNP complexes will enable numerous applications in therapeutic gene editing.

## Methods

**Materials.** SpCas9 and AsCpf1 were purchased from the MacroLab in UC Berkeley. Proteins were stored in 50 mM HEPES (4-(2-hydroxyethyl)-1-piperazineethanesulfonic acid) at pH 7.5 with 300 mM NaCl, 10% glycerol, and 100 μM tris(2-carboxyethyl) phosphine at −80 °C. PAsp(DET) polymer was synthesized using the following literature references[46,60,61]. Phusion High-fidelity DNA Polymerase was purchased from NEB (Ipswich, MA, USA). The MEGAscript T7 kit, the MEGAclear kit, the PageBlue solution, the propidium iodide, and the PureLink genomic DNA kit were purchased from Thermo Fisher (Waltham, MA, USA). Mini-PROTEAN TGX (Tris-Glycine eXtended) gels (4–20%) were purchased from Bio-Rad (Hercules, CA, USA). Dulbecco's modified Eagle's medium (DMEM), non-essential amino acids, penicillin–streptomycin, Dulbecco's phosphate-buffered saline (PBS), and 0.05% trypsin were purchased from Life Technologies (Carlsbad, CA, USA).

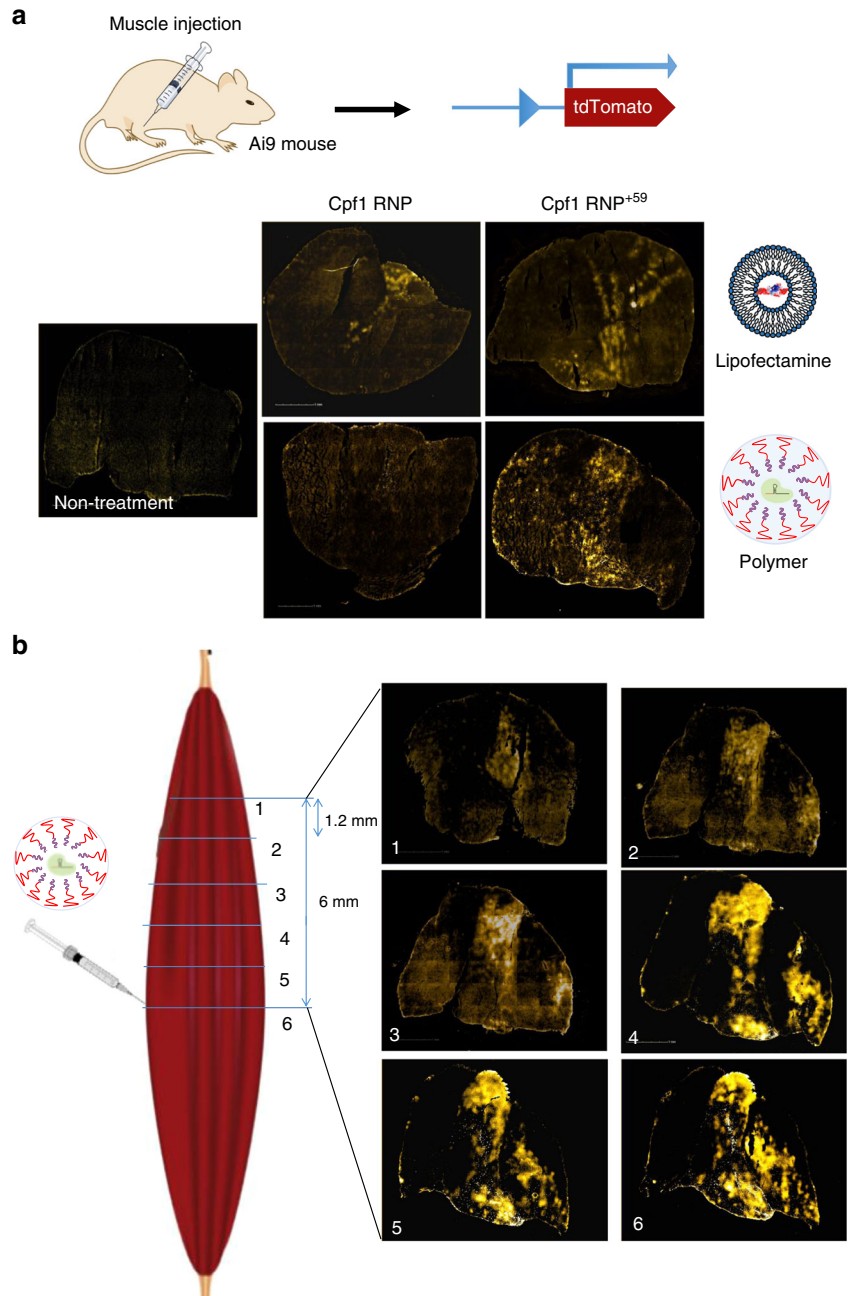

**Fig. 7** Cpf1 RNP$^{+59}$ enhances the delivery efficiency of Cpf1 RNP in vivo using polymer nanoparticle in Ai9 mice. **a** Fluorescence image of the entire gastrocnemius muscle cross-section that was injected with Cpf1 RNP. Cpf1 RNP efficiently induces gene deletions and robust expression of tdTomato in Ai9 mice after delivery with PAsp derivative polymer or Lipofectamine. Width of each image is 6.5 mm. **b** Serial fluorescence images of the entire gastrocnemius muscle injected with Cpf1 RNP, which was sectioned every 1.2 mm. Cpf1 RNP$^{+59}$ efficiently induces gene deletions in Ai9 mice after delivery with PAsp derivative polymer and shows robust gene editing along the muscle fiber. Width of each image is 6.5 mm

**Synthesis of gRNAs.** crRNAs and sgRNAs were purchased from IDT or synthesized using in vitro transcription. The sequences of gRNA are given in the Supplementary Data. The DNA template used for in vitro transcription was produced by overlapping PCR. Briefly, the forward primer and reverse primer (1 μM) were mixed with Phusion DNA Polymerase (NEB) and PCR amplification was conducted. The DNA template was purified with a PCR clean-up kit. The Megascript T7 RNA polymerase kit (Thermo Fisher) was used to make gRNAs. Polyacrylamide gel extraction was conducted to make sure uniform sized gRNAs were produced. The concentration of purified gRNA was determined with a Nanodrop 2000 (Thermo Fisher Scientific) and the final gRNA products were stored at −80 °C.

**HEK293T reporter cell lines.** HEK293T cells (293FT; Thermo Fisher Scientific), and derived cell lines, were grown in DMEM (Corning Cellgro, #10-013-CV) supplemented with 10% FBS (Seradigm #1500-500), and 100 U/ml penicillin and 100 μg/ml streptomycin (Pen-Strep; Life Technologies Gibco, #15140-122) at 37 °C with 5% $CO_2$. Monoclonal HEK-RT3-4 reporter cells, here referred to as "GFP-HEK," were generated through low-copy transduction of HEK293T human embryonic kidney cells with the amphotropic-pseudotyped retrovirus RT3GEPIR-sh.Ren.713[72], comprising an all-in-one Tet-On system enabling doxycycline-controlled EGFP expression. After puromycin (2.0 μg/ml) selection of transduced HEK239Ts, 36 clones were isolated and individually assessed for (i) growth characteristics, (ii) homogeneous morphology, (iii) sharp fluorescence peaks of doxycycline- (1 μg/ml) inducible EGFP expression (Guava EasyCyte, Millipore), (iv) relatively low fluorescence intensity to favor clones with single-copy reporter integration, and (v) high transfectability. HEK-RT3-4 cells are derived from the clone that performed best in these tests (Supplementary Fig. 7). BFP-HEK cells were generated by transfecting HEK cells with a BFP-containing lentivirus,

followed by fluorescence-activated cell sorting-based enrichment using the protocol published by Richardson et al.[45]. HEK cells were plated at a density of $5 \times 10^4$ cells per well in a 24-well plate, a day before transfection experiments.

**Isolation and culture and transfection of primary myoblasts.** All animal studies were performed according to authorized protocols and animals were treated following the policies of the Animal Care and Use Committee of the University of California, Berkeley. Primary myoblasts were obtained from Ai9 mice following the previously reported protocol from Conboy and co-workers[10] and Rando Yin et al.[11]. Briefly, the gastrocnemius and tibialis anterior muscles were harvested and incubated for 30 min in a digest medium (Thermo, #17703034). Myoblasts were cultured on collagen-coated plates with myoblast culture medium (Ham's F-10 Nutrient mixture, 20% FCS, 2.5 ng/ml basic fibroblast growth factor, penicillin–streptomycin), with replacement every 24 h. Cpf1 RNPs were transfected using electroporation. After transfection, cells were cultured for an additional 6 days, and the editing efficiency was detected by flow cytometry.

**Electroporation.** Cells ($2 \times 10^5$ cells after counting) were detached by Accutase and spun down at $600 \times 3\,g$ for 3 min and then washed with PBS. The Amaxa 96-well Shuttle system was used for electroporation following the manufacturer's protocol. Cpf1 RNP or Cas9 RNP with or without DNA (Cpf1: 50 pmol; crRNA: 50 pmol, Cas9: 50 pmol, sgRNA: 50 pmol, with or without ssDNA/ssRNA: 50 pmol) were prepared in 10 µl of electroporation buffer. After the electroporation, the cells were incubated at 37 °C in tissue culture plates with 500 µl of culture media. The culture media were changed 16 h after the electroporation.

**Lipofection.** Cpf1 RNP (50 pmol of Cpf1 and 50 pmol of crRNA) or Cas9 RNP (50 pmol of Cas9 and 50 pmol of sgRNA) was mixed with 2 µl of Lipofectamine 2000 in a 40 µl total volume in OptiMEM. The lipofection was conducted in OptiMEM media without serum, and an equal volume of 2× growth media was added to the cells after 4 h of lipofection to minimize cytotoxicity. The medium was changed 16 h after the lipofection and the cells were incubated for several days until further fluorescence analysis. ssDNA (1 µg) was added to Cpf1 RNP solution for ssDNA experiments. For the cell delivery quantification experiment, crRNAs labeled with Atto 495 were delivered with the same method. One hour after transfection, the medium was removed and cells were washed with PBS. Cells were cultured additional 3 h in culture medium before flow cytometry analysis.

**Polymer transfection.** Cpf1 RNP (50 pmol of Cpf1 and 50 pmol of crRNA) or Cas9 RNP (50 pmol of Cas9 and 50 pmol of sgRNA) was mixed with 10 µg of PAsp (DET) that was prepared in 20 µl of 20 mM HEPES buffer. Transfection was conducted in OptiMEM media without serum, and an equal volume of 2× growth media was added to the cells after 4 h of lipofection to minimize cytotoxicity. The medium was changed 16 h after the lipofection and the cells were incubated for several days and analyzed for fluorescence analysis.

**Serum stability assay.** One microgram of RNA was prepared in 10 µl of PBS containing 5% FBS and incubated at 37 °C for 0, 15, 30, or 60 min. Ten microliters of Gel Loading Buffer II (Thermo) was then added and heated at 70 °C for 5 min to denature RNA and proteins. RNA samples were loaded into a 4–20% TGX to detect intact RNA. Quantitative analysis was obtained by three independent experiments and band intensity quantification was conducted using the ImageLab software (Bio-Rad).

***SERPINA1* gene editing detection with droplet digital PCR.** HepG2 cells were transfected using the electroporation method described above. Four days after the electroporation, gDNA was extracted and digested using *Hin*dIII (NEB) at 37 °C for 1 h. HepG2 cells were purchased from the Cell Culture Facility at UC Berkeley (Original source: ATCC HB-8065). In each ddPCR reaction, 50–100 ng DNA, ddPCR primers, probes, 10 µl of ddPCR Supermix (No dUTP) (Bio-Rad), and nuclease-free water were added, to generate a 20 µl volume, and mixed well. Reaction mixtures, together with 70 µl QX200 Droplet generation oil, was loaded into the appropriate wells of an 8-channel droplet generation cartridge, according to the instruction manual. The cartridge was placed in the QX200™ Droplet Generator to generate the droplets, which were then transferred to a 96-well plate and amplified by standard PCR. Cycling conditions were: 95 °C for 5 min, 40 cycles of 94 °C for 30 s, and 58 °C for 1 min, followed by 98 °C for 10 min and final hold at 4 °C. The ramp rate was 2 °C/s. After thermal cycling, plates were placed in QX200™ Droplet Reader for data acquisition.

**In vitro cleavage gel.** Template GFP DNA that was synthesized via PCR (25 nM) was cleaved with Cpf1 RNP with and without extension of the crRNA (165 nM) in a 10 µl solution for 5, 15, and 60 min. Gel electrophoresis of the cleavage samples in Tris/Borate-sodium dodecyl sulfate buffer separated the template from the cleavage products. Nucleic acid staining was conducted with Sybr Safe and then the gel was imaged and the individual bands were quantified with ChemiDoc MP using the ImageLab software (Bio-Rad).

**Representative images.** Representative images and uncropped images can be found in Supplementary Figs. 8–14.

**Cpf1 plasmid transfection.** Cpf1 plasmid[18] was transfected into GFP-HEK cells ($2 \times 10^5$ cells) by mixing 1 µg of plasmid and 1 µl of Lipofectamine 2000 in OptiMEM. After 24 h transfection, electroporation was conducted for crRNA delivery.

**Flow cytometry analysis and fluorescence microscopy.** Flow cytometry (Attune Nxt Flow Cytometer, Thermo Fisher Scientific) was used to quantify the expression levels of BFP from BFP-HEK cells and of EGFP from GFP-HEK cells after transfection with potential editing reagents. The BFP-HEK cells were analyzed 7 days after transfection. The GFP-HEK cells were induced with doxycycline (1 µg/ml) 48 h after transfection. Fluorescence observation or quantification was conducted 48 h after the induction. Hoechst staining was conducted to visualize the nucleus, and fluorescence images were taken with a Zeiss Axioscope fluorescent microscope and analyzed with ImageJ. For flow cytometry, the cells were washed with PBS and detached by Accutase.

**Analysis of genome editing efficiency.** HDR efficiency was quantified by the restriction enzyme digestion of PCR-amplified target genes. Donor ssODN was designed to insert restriction enzyme sites, cleavable by *Cla*I, into the sequence that Cpf1 targets. The PCR amplicon of the BFP or GFP gene was incubated with the *Cla*I restriction enzyme (10 U) for 2 h at 37 °C. The products were analyzed by gel electrophoresis using a 4–20% Mini-PROTEAN TGX Gel (Bio-Rad) and stained with SYBR Green (Thermo Fisher). Individual band intensity was quantified using ImageLab and the HDR efficiency was calculated using the following equation: $(b + c) / (a + b + c) \times 100$ ($a$ = uncleaved PCR amplicon, $b$ and $c$ = the cleavage products).

**Gene editing in the muscle of Ai9 mice.** All animal studies were performed according to authorized protocols and animals were treated following the policies of the Animal Care and Use Committee of the University of California, Berkeley. Three groups of 4-week-old Ai9 mice (Jackson Laboratory, #007909) were used in this experiment, including control (no injection, $n = 1$), non-extended crRNA group ($n = 3$), and extended crRNA group ($n = 3$). Both male and female mice were chosen randomly for experiments and analysis was conducted in a non-blinded way. Cpf1 (200 pmol), crRNAs (200 pmol), and 40 µg PAsp derivative polymer were mixed and incubated for 2 min at room temperature. These nanoparticles were injected into gastrocnemius muscles (20 µl per muscle) using a 35 g WPI NanoFil syringe. Two weeks after the injection, the muscles were harvested, mounted in OCT, and flash frozen. Ten micrometer sections were cut using a cryostat. Entire gastrocnemius muscle was sectioned into hundreds of sections and observed throughout 6 mm range to understand how broad the effects of gene editing were. Pictures were taken by Opera Phenix High-Content Screening System with ×20 water objective and 10 areas were combined. Image J was applied to analyze the area of red fluorescence and total area. For Lipofectamine treatment, the non-extended crRNA group ($n = 1$) and extended crRNA group ($n = 1$) were injected with Cpf1 RNP with Lipofectamine: Cpf1 (200 pmol), crRNAs (200 pmol), and Lipofectamine 2000 (8 µl) were mixed and incubated for 5 min at room temperature.

**Data availability.** The data that support the findings of this study are available from the corresponding author upon reasonable request.

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

## Acknowledgements

We thank Jennifer Doudna for advice and support. We thank Kazunori Kataoka and Hyunjin Kim for advice regarding polymer. We also thank Mary West and the CIRM/QB3 Shared Stem Cell Facility/High-Throughput Screening Facility for providing flow cytometry support. C.F. is supported by a US National Institutes of Health K99/R00 Pathway to Independence Award (K99GM118909) from the National Institute of General Medical Sciences (NIGMS).

## Author contributions

H.M.P., H.L., and K.L. performed key in vivo experiments and primary cell culture experiments and wrote the manuscript. A.C., V.M., and A.R. performed in vitro experiments and DNA analysis. C.F. generated GFP-HEK cells and participated in discussion. F.J. provided structural insight and participated in discussion. H.C. conducted in vitro experiments. J.W. designed the HepG2 cell experiment and helped in writing the manuscript. K.L. planned and interpreted the data. N.M. and K.L. guided the project and wrote the manuscript.

## Additional information

**Competing interests:** K.L., H.M.P., and N.M. are co-founders of GenEdit Inc. The other authors declare no competing interests.

