## [Peer Review File · Nature Communications]

Reviewers' comments:

Reviewer #1 (Remarks to the Author):

In this manuscript entitled "Extension of the crRNA enhances Cpf1 gene editing in vitro and in vivo" by Liu et al., the authors find that 5' end extension of the Cpf1 crRNA leads to enhanced editing (both NHEJ and HDR). Whereas this is an interesting finding and holds translational potential, the work is somewhat preliminary and lacking of some depth.

Fig. 1. I do not find this figure very informative. If the point is to show the reader which end of the crRNA was extended, this can be conveyed in Figure 2A.

Fig. 2. The authors show that extending the 5' end of the Cpf1 crRNA leads to enhanced NHEJ (as judged by determining the % of GFP negative cells [not the greatest assay as it is based on a negative readout, but I am willing to accept this for now]). This point is demonstrated by electroporation of the AsCpf1 in their GFP-HEK cell line. What is the mechanism of this enhancement? The reason I ask is that in the beginning of the paper the authors claim that longer sgRNAs increases the activity of Cpf1 while later on in the paper, they argue that longer crRNA leads to more efficient delivery of the crRNA, but I doubt that can be the explanation for the enhanced effect here (or could it also be?). To address this, in vitro cleavage assay on a dsDNA template with all the different sgRNAs could be performed. As well, having a Cpf1-GFP fusion could help answer the question whether having a longer sgRNA leads to greater transfection efficiency of the RNPs. This brings me to the initial comparison between SpCas9 and Cpf1 (Fig 2b). The authors mention that SpCas9 can edit the genome more efficiently than Cpf1, could it simply be that transfection of SpCas9 is more efficient?

Fig. 2e. The authors demonstrate that chemical modifications to the 5' extended region does not impair editing efficiency. This is fine, but is there is no demonstrated advantage to having such modifications presented in this MS, so we are left wondering why these experiments are important.

Fig. 3 presents us with a very interesting result which the authors have not expanded on - in fact, there is now a white elephant in the room after looking at this data. The authors find that co-introduction of an ssODN with the AsCpf1 RNP by electroporation into the GFP-HEK cell line leads to almost 100% gfp-negative cells, regardless of the sequence of the oligo, a result that the authors indicate has not been previously reported. Yet no further work is performed in the MS documenting this striking effect, which to me would significantly elevate the impact of the current paper. I'd like to know if this effect is also seen by lipofection, with other crRNAs, documentation of the status of the GFP-negative alleles by sequencing to know whether or not oligo sequence has been introduced at the cut site, and whether this effect is also seen with SpCas9. Testing this on endogenous loci would also be critical. 100% editing is a big deal, this should be looked at closer.

Fig 3b. The editing efficiencies for HDR are reported to be ~ 17% (crRNA+4), and the editing efficiencies for NHEJ (Fig 3c) are almost 100%, hence we arrive at 118%. Is the oligo used in Fig 3b also rendering the cells GFP-negative? How many GFP copies are in this cell line (I'm assuming one, but has this been formally measured)? The explanation might be quite simple, but as written, this point is confusing and not well explained.

Figure 6. The authors indicate that extending the 5' end of the crRNA leads to enhanced delivery of the Cpf1 RNP. No direct measurement of this made in the MS and this information is critical to understanding why increasing the length of the crRNA affects editing efficiency. How do the authors reconcile the results in this figure with those in Fig 3, where the crRNA+4 and crRNA+9 are better than the crRNA +59.

Fig 4. The authors utilize a reporter system for which the schematic is presented in Fig 7A. This should be corrected so that figures can be followed in sequential order, and is required for understanding Figure 4. How does a non-specific ssDNA behave in this system? This is an important question, because up to this point, the presented results are all with one system. [I believe the BFP data in the Supplemental figure utilizes the same crRNA as used against GFP - authors, please correct me if I am wrong here and make this clearer in the text.]

Minor

Throughout the manuscript, the authors show bar graphs to present their editing efficiencies. It would be good if representative FACS plots were shown to help the reader visualize the effect observed. This could be presented as a supplemental figure. The same applies to Fig 3 where a gel showing the PCR digestion experiments would be adequate here.

Fig. 5 -Why is the Y axis in this figure labeled "NHEJ Efficiency (%)", whereas in all the other figures, it is "GFP-(%)". I believe GFP-negative is more reflective of what is actually being measured, not NHEJ.

Abstract: "that Cpf1 can be delivered adult mouse" should read "that Cpf1 can be delivered to adult mouse".

p.2: "that render themselves as attractive" should read "that render themselves as attractive".

p.2 Introduction 2nd paragraph: "Second, Cpf1 possess an innate" should read: "Second, Cpf1 possesses an innate"

Reviewer #2 (Remarks to the Author):

The authors showed that extension of the crRNA enhances Cpf1 gene editing, which is of great interest of the field. The major concerns are: 1. The authors tested only 4nt, 9nt and 59nt extension. Serial extensions should be tested to see how different extensions impact Cpf1 gene editing efficiency. Also, serial extensions should be performed in more than one crRNA to draw a solid conclusion. 2. In the figure 1, the effect of extension might be due to the negative charge change of RNP caused by extension because all of the experiment were done using RNP with either electroporation or lipid transfection, both are affected by RNP charge. The authors will need to express the crRNA in a vector to show whether extension can enhance Cpf1 functionality. 3. Authors claimed that extension of crRNA can increase gene editing efficiency to a level comparable to the commonly used SpCas9/sgRNA system based on only one case, which should not lead to the conclusion.

Reviewers' comments:

Reviewer #1 Questions

General comment

In this manuscript entitled "Extension of the crRNA enhances Cpf1 gene editing in vitro and in vivo" by Liu et al., the authors find that 5' end extension of the Cpf1 crRNA leads to enhanced editing (both NHEJ and HDR). Whereas this is an interesting finding and holds translational potential, the work is somewhat preliminary and lacking of some depth.

Answer. We thank the Reviewer for their thoughtful comment.

Question 1. Fig. 1. I do not find this figure very informative. If the point is to show the reader which end of the crRNA was extended, this can be conveyed in Figure 2A.

Answer 1. We thank the Reviewer for their comment. We have redrawn Figure 1 and it now conveys the major points of the paper in a detailed manner.

Question 2. Fig. 2. The authors show that extending the 5' end of the Cpf1 crRNA leads to enhanced NHEJ (as judged by determining the % of GFP negative cells [not the greatest assay as it is based on a negative readout, but I am willing to accept this for now]. This point is demonstrated by electroporation of the AsCpf1 in their GFP-HEK cell line. What is the mechanism of this enhancement? The reason I ask is that in the beginning of the paper the authors claim that longer sgRNAs increases the activity of Cpf1 while later on in the paper, they argue that longer crRNA leads to more efficient delivery of the crRNA, but I doubt that can be the explanation for the enhanced effect here (or could it also be?). To address this, in vitro cleavage assay on a dsDNA template with all the different sgRNAs could be performed. As well, having a Cpf1-GFP fusion could help answer the question whether having a longer sgRNA leads to greater transfection efficiency of the RNPs.

Answer 2. We thank the Reviewer for their helpful comment, and have performed multiple experiments to determine the mechanism by which the crRNA extension enhances gene editing activity. We have performed in vitro cleavage assays with the extended crRNA and did not observe any increase in cleavage rate versus the wild type crRNA. In addition, we conducted an experiment in which Cpf1 was delivered via plasmid DNA, with crRNAs that had extensions, and in this experiment, we did not observe an increase in gene editing efficiency.

In addition, we have also fluorescently labeled the crRNA to determine if its uptake is enhanced because of the 5' extension. We chose to fluorescently label the crRNA with a fluorescent dye because expression and purification of a Cpf1-GFP fusion protein is not straightforward due to its high molecular weight. The fluorescently labeled Cpf1 RNPs were delivered using electroporation and lipofectamine. Electroporation was very efficient at delivering even wild type Cpf1 RNP (no extension) and 95% of the electroporated cells were positive for all crRNAs, regardless of crRNA length. In contrast, the crRNA extension increased the cell uptake of Cpf1 RNP delivered via Lipofectamine. This experiment suggests that the extension of the crRNA increases its interaction with lipofectamine, and provides a mechanism by which the crRNA extension enhances Cpf1 RNP delivery into cells via lipofectamine.

We have changed the text of the manuscript to describe these new experiments. It now states on page 15:

“We performed experiments to determine the mechanism by which the 5' extended crRNAs enhanced the gene editing efficiency of the Cpf1 RNP. First, we tested whether the extended crRNAs enhance the inherent nuclease activity of Cpf1, using an in vitro DNA cleavage assay. We observed no activity difference between the three crRNAs tested, wild type crRNA, crRNA⁺⁹, and crRNA⁺⁵⁹ with 15 min and 60 min incubation time (Supplementary Figure 4 and 5). crRNA⁺⁵⁹ even had slower DNA cleavage than wild type crRNA when the incubation time was only 5 min. This result suggests that crRNA extension does not enhance the nuclease activity of the Cpf1 RNP in tube. We also investigated if 5' extended crRNAs enhanced the gene editing activity if the Cpf1 was delivered by plasmid rather than as an RNP. Cpf1 plasmid was transfected 24 hours prior to electroporation of the crRNAs and the gene editing activity was determined. Extended crRNAs showed no improvement in gene editing efficiency when the Cpf1 was produced from plasmids (Supplementary Figure 6).

Finally, we labeled the crRNAs with a fluorescence dye to determine if the extended crRNAs had enhanced uptake in cells after delivery via either electroporation or lipofectamine. Electroporation of the Cpf1 RNPs resulted in above 90% of the cells being positive for the dye-crRNA and showed highly efficient delivery regardless of the crRNA length. On the other hand, the delivery efficiency of Cpf1 RNP with lipofectamine was dependent on the length of the crRNA, and extended crRNAs were delivered into HEK 293T cells more efficiently than wild type crRNA (Figure 6d). The net charge of a macromolecule is a critical parameter for efficient interaction with lipofectamine. Extension of the crRNA significantly increases the net negative charge of the Cpf1 RNP, which should result in more efficient interaction with lipofectamine and efficient delivery into cells.”

Question 3. The authors mention that SpCas9 can edit the genome more efficiently than Cpf1, could it simply be that transfection of SpCas9 is more efficient?

Answer 3. Yes, it is possible that the transfection efficiency of the SpCas9 is more efficient than Cpf1. We have deleted the sentence in the text that compares the gene editing efficiency of SpCas9 with Cpf1.

Question 4. Fig. 2e. The authors demonstrate that chemical modifications to the 5' extended region does not impair editing efficiency. This is fine, but there is no demonstrated advantage to having such modifications presented in this MS, so we are left wondering why these experiments are important.

Answer 4. We thank the Reviewer for their constructive comment. We have performed serum stability experiments with crRNAs that had a 5' extension, in which the extended bases were modified with phosphorothioate linkages. We have been able to demonstrate that the phosphorothioate modified crRNAs are more stable in serum than unmodified crRNA. Backbone modified crRNA⁺⁹ (crRNA^{+9S}) was 40% intact after 15 minutes in diluted serum, whereas crRNAs without backbone modification (crRNA⁺⁹ or crRNA) were completely hydrolyzed. In addition, we have done lipofectamine transfection experiments with the modified crRNA in GFP-HEK cells and observed that it had a higher transfection efficiency than wild type extended crRNA, presumably due to greater protection from nucleases in cells. Importantly, modifying unextended crRNA with phosphorothioate linkages completely destroys its activity, whereas the 5' crRNA extension is still active after modification with phosphorothioate linkages. The 5' crRNA

extension thus provides a powerful methodology for introducing chemical modifications onto the crRNA.

We have changed the text of the manuscript to describe these new experiments. It now states on page 12:

“A key benefit of using chemically modified crRNAs is that they are more stable to hydrolysis by serum nucleases. Therefore, the serum stability of the 5' chemically modified crRNAs was investigated. 5' chemically modified crRNAs were incubated in diluted fetal bovine serum and their degradation was analyzed via gel electrophoresis. Figure 5c and 5d show that unmodified crRNAs rapidly degrade in serum, whereas crRNA^{+9S}, which contains a phosphorothioate backbone, is significantly more stable to hydrolysis in serum. In addition, we investigated if 5' modified crRNAs could enhance the ability of lipofectamine to transfect Cpf1 RNP, due to its ability to protect the crRNA from nucleases in cells and in serum. Cpf1 with crRNA^{+9S} was more efficient at editing genes in cells than crRNA⁺⁹ by 40%, and this suggests that 5' crRNA chemical modifications, enabled by 5' crRNA extension, will have numerous applications in gene editing (Figure 5e).”

Question 5. Fig. 3 presents us with a very interesting result, which the authors have not expanded on - in fact, there is now a white elephant in the room after looking at this data. The authors find that co-introduction of an ssODN with the AsCpf1 RNP by electroporation into the GFP-HEK cell line leads to almost 100% gfp-negative cells, regardless of the sequence of the oligo, a result that the authors indicate has not been previously reported. Yet no further work is performed in the MS documenting this striking effect, which to me would significantly elevate the impact of the current paper. I'd like to know if this effect is also seen by lipofection, with other crRNAs, documentation of the status of the GFP-negative alleles by sequencing to know whether or not oligo sequence has been introduced at the cut site, and whether this effect is also seen with SpCas9. Testing this on endogenous loci would also be critical. 100% editing is a big deal, this should be looked at closer.

Answer 5. We thank the Reviewer for their thoughtful comment. We have performed additional gene editing studies with exogenous ssDNA and Cpf1 gene editing, using primary myoblasts obtained from the Ai9 mouse. We observed that ssDNA enhanced the gene editing efficiency of Cpf1 RNP in primary myoblasts and suggests that this method will be broadly applicable for enhancing gene editing (Figure 4c).

In order to test whether there is integration of ssDNA to the target cut site, we have done PCR amplification of genomic DNA 3 days after transfection with primer sets with one primer binding ssDNA and the other primer binding the target genomic DNA region. PCR analysis showed no amplification of ssDNA sequences in the genomic GFP target sequence and only the control GFP PCR showed bands. This experiment demonstrates that ssDNA integration into the genome is not occurring at high frequency (Supplementary Figure 8).

We further tested whether the ssDNA effect is observed with lipofectamine transfection of Cpf1 RNP in GFP-HEK cells. We observed that there is no gene editing enhancement effect when Cpf1 RNP + ssDNA was delivered with lipofectamine (Supplementary Figure 1). We found a methods paper published by IDT, which demonstrates that ssDNA is an electroporation enhancer for Cas9 RNP, and they also did not observe an enhancement with lipofection^{1,2}. We have re-written the manuscript to more clearly describe the ssODN effect, and cite relevant publications on the ssODN effect with SpCas9. It now states on page 7:

“ssDNA can augment the editing efficiency of SpCas9^{47,48}, delivered via

electroporation and our results demonstrate that ssDNA can augment editing with AsCpf1 as well. Additionally, the activity enhancement with 5'-end extension was synergistic with exogenous ssDNA and collectively the gene editing they induced was close to a 100%, which is a level that had not been reported previously. We also tested whether ssDNA could increase the gene editing efficiency of Cpf1 after delivery into cells via lipofectamine. Supplementary Figure 1 shows that the addition of ssDNA does not enhance the Cpf1 gene editing efficiency, if lipofectamine is used as the delivery method. This result limits the usage of ssDNA as an enhancer of gene editing to the electroporation method.

We further investigated whether single-stranded RNA (ssRNA) can enhance the gene editing efficiency of AsCpf1. ssDNA is potentially problematic to use for enhancing AsCpf1 gene editing activity because ssDNA can potentially integrate into the genome and cause genomic damage, in contrast, ssRNA cannot integrate into the genome, and would be much safer to use. GFP-HEK cells were electroporated with Cpf1 RNP and two different ssRNAs (9nt and 100nt) and the resulting levels of gene editing were determined. Two 100nt ssRNAs with slight variation both dramatically increased the gene editing efficiency of Cpf1, resulting in a 2-fold improvement, whereas the 9nt ssRNA induced only a 10% enhancement in NHEJ efficiency (Figure 3e). These results suggest that 100nt ssRNA can be potentially used as a gene editing enhancer for Cpf1 RNP, and provides a safe alternative to ssDNA.”

Question 6. Fig 3b. The editing efficiencies for HDR are reported to be ~ 17% (crRNA+4), and the editing efficiencies for NHEJ (Fig 3c) are almost 100%, hence we arrive at 118%. Is the oligo used in Fig 3b also rendering the cells GFP-negative? How many GFP copies are in this cell line (I'm assuming one, but has this been formally measured)? The explanation might be quite simple, but as written, this point is confusing and not well explained.

Answer 6. We thank the reviewer for their comment. The ssODN that was used for the HDR experiment had a sequence that inserts a restriction enzyme site and causes knock-out of GFP as a consequence. Therefore, the GFP negative population includes both the HDR and the NHEJ population. We have revised the manuscript to explain this clearly. It now states in page 7:

“We also examined whether the 5'-extension could increase HDR rates in addition to NHEJ levels. The AsCpf1 RNPs with crRNA containing various extensions were introduced into GFP-HEK cells together with a single-stranded oligonucleotide donor (ssODN) (Figure 3a). GFP negative cells included both frame-shift mutation caused by HDR and indel mutations caused by NHEJ. HDR rates were quantified using a restriction enzyme digestion assay³⁻⁶. A 2-fold improvement in HDR was observed for both the 4 and 9 nt extended crRNAs (17% HDR frequency for crRNA⁺⁴ and 18% HDR frequency for crRNA⁺⁹ versus 9% for control crRNA in Figure 3b).”

We generated the GFP-HEK reporter cell line using low viral concentrations for the transfection, to minimize the possibility of multiple copy integration. We collected a clone that has the lowest GFP fluorescence intensity compared to other tested clones, to pick a monoclonal cell line most likely to contain only a single GFP integration (single GFP copy), we have not specifically measured the GFP copy number by Southern blotting. To illustrate the cell line generation and selection process more clearly, we added an additional supplementary figure (Supplementary Fig. 7) showing the GFP fluorescence profiles of various tested GFP-HEK monoclonal cell lines. Additionally, we also extended the methods section that describes the GFP-HEK cell line generation (Methods).

Question 7. Figure 6. The authors indicate that extending the 5' end of the crRNA leads to enhanced delivery of the Cpf1 RNP. No direct measurement of this made in the MS and this information is critical to understanding why increasing the length of the crRNA affects editing efficiency. How do the authors reconcile the results in this figure with those in Fig 3, where the crRNA+4 and crRNA+9 are better than the crRNA +59.

Answer 7. We have conducted multiple experiments to understand the mechanism of how the crRNA extension improves the gene editing efficiency. We tested 15nt and 25nt extensions in GFP-HEK cells and repeated the myoblast gene editing experiments. The results from these cell experiments showed that crRNA extensions 4-25 nucleotides in length all significantly enhanced gene editing efficiency of Cpf1 RNP. In the ai9 myoblast experiment, crRNA⁺⁵⁹ still showed comparable levels of gene editing to crRNA⁺⁹. However, we still cannot pinpoint the optimal length or exact mechanism by which the crRNA extension enhances the activity of Cpf1 after electroporation. Although we are very interested in determining this, we feel that these mechanistic experiments are beyond the scope of this paper.

Question 8. Fig 4. The authors utilize a reporter system for which the schematic is presented in Fig 7A. This should be corrected so that figures can be followed in sequential order, and is required for understanding Figure 4. How does a non-specific ssDNA behave in this system? This is an important question, because up to this point, the presented results are all with one system. [I believe the BFP data in the Supplemental figure utilizes the same crRNA as used against GFP - authors, please correct me if I am wrong here and make this clearer in the text.]

Answer 8. We thank the reviewer for their constructive comment. We have changed Figures 4 and 7a so that the Figures can be followed in a sequential order. Regarding the question about ssDNA, we have conducted additional experiments on primary myoblasts from the ai9 mouse (Figure 4c) to answer this question. ssDNA with no significant sequence homology to the target genomic DNA was added to the Cpf1 RNP and electroporated into ai9 cells, and we observed a statistically significant enhancement in the RFP+ population, which is consistent with the effects seen on HEK293T cells.

Minor

Question 9. Throughout the manuscript, the authors show bar graphs to present their editing efficiencies. It would be good if representative FACS plots were shown to help the reader visualize the effect observed. This could be presented as a supplemental figure. The same applies to Fig 3 where a gel showing the PCR digestion experiments would be adequate here.

Answer 9. We thank the reviewer for their helpful comment. We have included representative FACS plots of the data, where relevant, to help the reader visualize the data. We have also included a gel in the Supplementary Information that shows a representative gel image of the restriction enzyme assay used to quantify HDR in Figure 2b.

Question 10. Fig. 5 -Why is the Y axis in this figure labeled "NHEJ Efficiency (%)" , whereas in all the other figures, it is "GFP-(%)". I believe GFP-negative is more reflective

of what is actually being measured, not NHEJ.

Answer 10. We have changed the Y axis in Figure 5 from NHEJ efficiency (%) to GFP- (%).

Question 11. Abstract: "that Cpf1 can be delivered adult mouse" should read "that Cpf1 can be delivered to adult mouse".

Answer 11. We have corrected this typo and the revised the abstract.

Question 12. p.2: "that render themselves as attractive" should read "that render themselves as attractive".

Answer 12. We have corrected this typo on p.2.

Question 13. p.2 Introduction 2nd paragraph: "Second, Cpf1 possess an innate" should read: "Second, Cpf1 possesses an innate"

Answer 13. We have corrected the typo on p.2 Introduction 2nd paragraph.

Reviewer #2

General comments

The authors showed that extension of the crRNA enhances Cpf1 gene editing, which is of great interest of the field.

Answer. We thank the Reviewer for their kind comment.

Question 1. The authors tested only 4nt, 9nt and 59nt extension. Serial extensions should be tested to see how different extensions impact Cpf1 gene editing efficiency. Also, serial extensions should be performed in more than one crRNA to draw a solid conclusion.

Answer 1. We thank the Reviewer for their thoughtful comment. We have conducted additional Cpf1 RNP electroporation experiments with crRNA extensions that were 15nt and 25nt in length in GFP-HEK cells. The results from the GFP-HEK cell experiments show that extensions ranging from 4nt to 25nt all significantly enhanced the gene editing efficiency of Cpf1, after electroporation (Figure 2c).

We have revised the manuscript to include these new experiments. It now states on page 5:

“To determine if crRNA 5'-extensions affect Cpf1 gene editing, we compared the activities of crRNAs with 5'-extensions of various lengths using our GFP-HEK reporter system. GFP-targeting crRNAs with 5'-end extensions of 4, 9, 15, 25, and 59 nucleotide (nt) were introduced into GFP-HEK cells by electroporation as an RNP complex with AsCpf1. The sequences for the 4 to 25 nucleotide extensions were scrambled, and the 59 nt extension consisted of the AsCpf1 pre-crRNA^{7,8} preceded by a scrambled RNA sequence with no homology to human genome sequence. The crRNAs with the 4 to 25 nt 5'-extensions all exhibited dramatically increased gene editing over the crRNA with no extension. Cells electroporated with the: unextended crRNA were 30% GFP negative (crRNA), 4 to 25 nucleotide extended crRNA were 55 to 60% GFP negative and 59 nucleotide extended crRNA were 37% GFP negative (crRNA⁺⁵⁹) (Figure 2c).”

Question 2. In the figure 1, the effect of extension might be due to the negative charge change of RNP caused by extension because all of the experiment were done using RNP with either electroporation or lipid transfection, both are affected by RNP charge. The authors will need to express the crRNA in a vector to show whether extension can enhance Cpf1 functionality.

Answer 2. We thank the Reviewer for their thoughtful comment and have performed additional experiments to examine this point. We have performed in vitro cleavage assays with the extended crRNAs and did not observe an increase in cleavage rate versus the wild type crRNA, and the crRNA extension is therefore not enhancing the Cpf1 nuclease activity. In addition, we delivered Cpf1 encoding plasmid DNA and crRNAs with and without extension, and measured the gene editing efficiency in GFP-HEK cells. In these experiments, we did not observe an increase in gene editing efficiency with extended crRNAs, when Cpf1 was delivered via plasmid DNA. Thus the enhancement in Cpf1 transfection, observed with the crRNA extension, is restricted to the RNP format, suggesting that it is related to the negative charge change and delivery efficiency. The correlation of negative charge and delivery efficiency with lipofectamine was further investigated with fluorescently labeled crRNAs. Cpf1 RNPs with fluorescently labeled crRNA with different lengths were delivered with lipofectamine. We observed a

significant correlation between the fluorescence intensity in cells with the length of the crRNA that was delivered.

We have described these new experiments on page 15, it now states:

“We performed experiments to determine the mechanism by which the 5' extended crRNAs enhanced the gene editing efficiency of the Cpf1 RNP. First, we tested whether the extended crRNAs enhance the inherent nuclease activity of Cpf1, using an in vitro DNA cleavage assay. We observed no activity difference between the three crRNAs tested, wild type crRNA, crRNA⁺⁹, and crRNA⁺⁵⁹ with 15 min and 60 min incubation time (Supplementary Figure 4 and 5). crRNA⁺⁵⁹ even had slower DNA cleavage than wild type crRNA when incubation was only 5 min. This result suggests that crRNA extension does not enhance the nuclease activity of the Cpf1 RNP in tube. We also investigated if 5' extended crRNAs enhanced the gene editing activity if the Cpf1 was delivered by plasmid rather than as an RNP. Cpf1 plasmid was transfected 24 hours prior to electroporation of the crRNAs and the gene editing activity was determined. Extended crRNAs showed no improvement in gene editing efficiency when the Cpf1 was produced from plasmids (Supplementary Figure 6).

Finally, we labeled the crRNAs with a fluorescence dye to determine if the extended crRNAs had enhanced uptake in cells after delivery via either electroporation or lipofectamine. Electroporation of the Cpf1 RNPs resulted in above 90% of the cells being positive for the dye-crRNA and showed highly efficient delivery regardless of the crRNA length. On the other hand, delivery efficiency of Cpf1 RNP with lipofectamine was dependent on the length of the crRNA, and extended crRNAs were delivered into HEK 293T cells more efficiently than wild type crRNA (Figure 6d). The net charge of a macromolecule is a critical parameter for efficient interaction with lipofectamine. Extension of the crRNA significantly increases the net negative charge of the Cpf1 RNP, which should result in more efficient interaction with lipofectamine and efficient delivery into cells.”

Question 3. Authors claimed that extension of crRNA can increase gene editing efficiency to a level comparable to the commonly used SpCa9/sgRNA system based on only one case, which should not lead to the conclusion.

Answer 3. We thank the Reviewer for their constructive comment. We have removed this sentence from the text of the manuscript.

References

1. Richardson, C. D., Ray, G. J., Bray, N. L. & Corn, J. E. Non-homologous DNA increases gene disruption efficiency by altering DNA repair outcomes. *Nat. Commun.* **7**, 1–7 (2016).
2. Jacobi, A. M. *et al.* Simplified CRISPR tools for efficient genome editing and streamlined protocols for their delivery into mammalian cells and mouse zygotes. *Methods* **121–122**, 16–28 (2017).
3. Byrne, S. M., Mali, P. & Church, G. M. *Genome editing in human stem cells. Methods in Enzymology* **546**, (Elsevier Inc., 2014).
4. DeWitt, M. A., Corn, J. E. & Carroll, D. Genome editing via delivery of Cas9 ribonucleoprotein. *Methods* **121–122**, 9–15 (2017).
5. Lin, S., Staahl, B., Alla, R. K. & Doudna, J. a. Enhanced homology-directed human genome engineering by controlled timing of CRISPR/Cas9 delivery. *Elife* **3**, 1–13 (2014).
6. Schumann, K. *et al.* Generation of knock-in primary human T cells using Cas9 ribonucleoproteins. *Proc. Natl. Acad. Sci.* **112**, 10437–10442 (2015).
7. Zetsche, B. *et al.* Cpf1 Is a Single RNA-Guided Endonuclease of a Class 2 CRISPR-Cas System. *Cell* **163**, 759–771 (2015).
8. Zetsche, B. *et al.* Multiplex gene editing by CRISPR–Cpf1 using a single crRNA array. *Nat. Biotechnol.* **35**, 31–34 (2017).

REVIEWERS' COMMENTS:

Reviewer #1 (Remarks to the Author):

The authors have addressed my concerns in a scholarly manner.

Reviewer #2 (Remarks to the Author):

The authors have solved all my concerns. I highly recommend the manuscript to be published right away.